# Experimental Research on Dynamic Behavior of Stiffened Plates under Drop-Weight Impacts of a Wedge Impactor

**DOI:** 10.3390/ma16083128

**Published:** 2023-04-15

**Authors:** Rong Yu, Wei Luo, Hao Chen, Jingxi Liu

**Affiliations:** 1School of Mechanical Technology, Wuxi Institute of Technology, Wuxi 214121, China; 2China Ship Development and Design Centre, Wuhan 430064, China; 3School of Naval Architecture and Ocean Engineering, Huazhong University of Science and Technology, Wuhan 430074, China; 4Hubei Key Laboratory of Naval Architecture and Ocean Engineering Hydrodynamics (HUST), Wuhan 430074, China

**Keywords:** stiffened plate, mild steel, wedge impactor, drop-weight impact test

## Abstract

Deck structures subjected to drop-weight low-velocity impact are critical safety elements for ships and offshore structures. Therefore, the aim of the present study is to propose experimental research on dynamic responses of deck structures composed of stiffened plates subjected to drop-weight impact of a wedge impactor. The first step was to fabricate a conventional stiffened plate specimen and a strengthened stiffened plate specimen, as well as a drop-weight impact tower. Then, drop-weight impact tests were carried out. Test results show that local deformation and fracture occurred in the impact area. A sharp wedge impactor caused premature fracture, even under relative low impact energy; the permanent lateral deformation of the stiffened plate was reduced by 20–26% by the strengthening effect of a strengthening stiffer; residual stress and the stress concentration of the cross-joint caused by welding may cause undesired brittle fracture. The present investigation provides useful insight for improving the crashworthiness design of the deck structure of ships and offshore structures.

## 1. Introduction

Ships and offshore structures often suffer low-velocity impact of falling objects such as containers and drill collars in accidents during deck-lifting operations. Especially for drilling ships, research ships and offshore structures, falling object accidents occur frequently. The impact of falling objects on the deck structure can cause economic losses and endanger the safety of personnel [1,2]. Nowadays, growing demands to exploit the ocean have led to an explosion in the number of research ships, engineering ships and offshore platforms. Thus, the probability of falling-object impact accidents are greatly increased. Therefore, it is worth paying special attention to the crashworthiness of deck structures under falling-object impact.

Experimental research is the basic method used to test the crashworthiness of ship structures subjected to collisions, as well as drop-weight impact. Generally, there are four typical methods used to estimate dynamic responses under low-velocity impact, including the empirical method, experimental method, numerical simulation method and analytical method. Currently, the empirical method is seldom used. Dynamic impact responses of plates, beams and cylinders have been investigated by the analytical method [3,4]. Due to its acceptable accuracy and efficiency, the analytical method has been widely used for rapid prediction during the early design stage. The numerical method has been widely used in the past decade [5,6]. For a given structure, the numerical method has shown powerful capabilities. Nevertheless, numerical simulation requires the consumption of considerable computing resources, as well as substantial modelling efforts, with challenges associated with mesh convergence, material failure criterion definition, strain-rate sensitivity, etc. In the analytical method, deformation and failure modes need to be obtained by observation through experimental tests. In numerical simulation, experimental results are often used to verify the accuracy of numerical simulation results. In other words, experimental investigations provide a benchmark for structural crashworthiness research [7].

A considerable amount of research has been proposed to investigate the dynamic performance of deck structures under falling-object impact. Ellis N [8] conducted an experimental study of responses of deck structures under striking of wellhead modules, wherein the wellhead modules struck the deck structure at a velocity of 17.32 m/s. Paik JK [9] proposed an empirical formula to predict energy absorption of deck structures under the striking of a wedge indenter. Obaid YF [10] proposed a Fortran program to predict responses of the deck under the impact of a falling drill collar based on an empirical method. Colwill AD and Ahilan AV [11] determined the crashworthiness of the deck of an offshore platform based on experimental test results; a computer program was introduced to establish the relationship between impact velocity and falling probability, resulting in the establishment of a reliability model. Sun LP [12] proposed non-liner finite-element analysis of the responses of deck structures under falling-object impact and introduced a simplified method by simplifying the stiffened plate as an equivalent plate for rapid prediction. Andreas B [13] proposed numerical simulations on the dynamic responses of deck structures under striking of a falling container based on ABAQUS. Zhou S, et al. [14] presented numerical simulations of the responses of a deck structure subjected to the impact of tubes based on LS-DYNA; the nonlinear relationship between the impact angle and the impact result were established using a BP neural network. Kim BJ [15] carried out experimental investigations on the mechanical responses of the structure of liquid natural gas container; a gun-type impact test machine was used to simulate free fall of a striking object from a height of 27 m. Seo JK, et al. [16] carried out an experimental test of the structural behavior of aluminum helicopter deck structures of offshore platforms under impact loads. Zhu L, et al. [17] determined the dynamic plastic response of rectangular plates subjected to rectangular rigid-mass impact by a drop-weight impact tower. Yang L and Wang DY [18] investigated the dynamic responses of a stiffened steel panel under the dropping impact of a rigid parabolically shaped falling object. Chen BQ et al. [19] proposed a wedge-shaped impactor to simulate a bulbous bow and conducted an experimental investigation on the responses of a double-hull ship side structure under the impact of a wedge impactor. Zhou H, et al. [20] determined the dynamic responses of deck structures under the impact of a falling sharp object to simulate the corner of a falling container and established a similarity law between a small-scale model and a full-scale prototype. Compared to conventional stiffened plates, novel structures including sandwich panels have the advantage of loading efficiency; therefore, the impact responses of novel structures have recently become a research hotspot. Zhu L, et al. [21] conducted a drop-tower impact test on the dynamic behavior of porous aluminum foam-filled sandwich panels under the impact of a hemispherical impactor. Li YG, et al. [22] determined the dynamic responses of PVC foam-filled sandwich structures subjected to a rigid wedge impactor. Wang ZP et al. [23] investigated the structural deformation of plates with hat-shaped stiffeners subjected to a hemispherical impactor. The impact of floating ice on ships and offshore structures has also attracted special attention. Zhu L, et al. [24] established a curved impact test rail, investigated the dynamic responses of rectangular plates under the impact of a wedge-ice impactor and compared elastic–plastic responses of plates under the impact of a ridged wedge striker and a brittle ice-wedge striker; the plastic responses of aluminum foam-filled sandwich panels under wedge-ice impact were also investigated [25]. Yu TQ, et al. [26] investigated the dynamic responses of stiffened panels under the impact of a brittle wedge-ice impactor.

The abovementioned investigations mainly focused on the responses of deck structures under axial striking of cylindrical members such as tubes and drill collars, and little attention has been paid to the dynamic responses of typical deck structures composed of cross-stiffened plates with consideration of an impactor with sharp edges. In the present investigation, we conducted experimental research on the dynamic responses of deck structures subjected to the falling impact of a wedge impactor. The first step was to establish a drop-weight impact test tower; then, a wedge impact was proposed to simulate the impact of a falling container, and two types of stiffened deck structure specimens were proposed: a conventional stiffened plate and a strengthened stiffened plate. In the following sections, we present our experimental investigations on impact responses with consideration of the influence of the impact location and the structural strengthening effect.

## 2. Impact Test Setup

A drop-weight impact test tower was established for low-velocity impact. As shown in Figure 1, the impact test tower consists of a rail, a drop-weight mass, a force transducer, a laser-displacement transducer and an impactor. During the impact procedure, the test specimen was fixed to the basement at the bottom of the tower by dozens of bolts to prevent the specimen from bounding or sliding away. The impactor was guided by two vertical rails and consisted of a rigid impactor, a force transducer and a mass. During the impact process, the initial impact energy was achieved by adjusting the drop height of the impactor and the mass property of the impactor. A piezoelectric force transducer (code number L1100–909543) provided by Xiyuan Electronic Technology Co., Ltd. (Yangzhou, China) was installed on the top side of the impactor. Impact force–time history was determined by the piezoelectric force transducer with a sampling frequency of 40 kHz.

In order to simulate the falling impact of a TEU (twenty feet equivalent unit) container, the total mass of the drop weight was set to 362.2 kg in the present research. The maximum gross weight of a twenty-feet equivalent unit container is about 23,000 kg; therefore, in order to simulate the impact of a dropping TEU container, the mass property of the impactor in the impact test tower was calculated according to the scaling law: *m* = *M*/*λ*^3^, where *m* represents the mass of the impactor in the impact tower, *M* represents the mass of the TEU container and *λ* represents the scaling factor. In the present research, *λ* = 4; thus, the mass properties were set approximately equal to 359 kg. In the present case, the mass of the impactor was 362.2 kg, with acceptable accuracy.

A laser displacement transducer fixed to the ground (code number HG–C 1400) fabricated by Panasonic Co., Ltd. (Kasugai, Japan) was used to measure the vertical displacement of the impactor. It makes sense that the initial impact velocity can be calculated based on the differential of displacement. Furthermore, energy absorption during the impact procedure was calculated by trapezoidal numerical integration of the force–displacement curve.

A wedge impactor derived from a container was used to simulate the falling impact of a container. As shown in Figure 2, since the sharp edge of a dropping container would cause deck damage in the case of falling down, a wedge impactor was used to simulate the edge of the container. The impactor was fabricated with abrasive steel, and a heat treatment process ensured its surface hardness.

## 3. Test Specimens

### 3.1. Specimen Design and Fabrication

Two types of rectangular steel plate specimens were used in the present investigation. Rectangular mild steel plates strengthened by stiffeners derived from the deck structure of an offshore platform were used in the present research. As shown in Figure 3a, a conventional rectangular plate with dimensions of 700 × 525 mm and a thickness of 3.9 mm was stiffened by three L-shaped stiffeners (L 59 × 15 × 3.9 mm) on the bottom side. A strengthened stiffened plate specimen was also used. As shown in Figure 3b, a strengthening stiffener perpendicular to the other stiffeners was welded on the bottom side of the plate. Photographs of the two specimens are shown in Figure 4. In order to simulate the fixed boundary condition of the plate, the plate was bolted to the basement, as shown in Figure 4c. Meanwhile, the three stiffeners were welded to stringers at the two ends, and the two stringers were also bolted to the basement to simulate the fixed boundary, as shown in Figure 4d.

### 3.2. Mechanical Properties

Both a quasistatic tensile test and a dynamic tensile test of the mild steel were carried out at room temperature. Under the guidance of the Chinese metallic materials tensile test standard GB/T 228.1 [27], dog-bone-shaped test specimens were cut from the mild steel plate by wire electrical discharge, as shown in Figure 5. The quasistatic test specimen and the dynamic test specimen have the same thickness of 3.9 mm, which is equal to the thickness of the panel of the rectangular dropping weight impact test specimens. For the quasistatic tensile test specimen, the length of a straight-line segment located in the middle of the specimen was set to 85 mm, and the width was set to 12.5 mm. Meanwhile, for the dynamic tensile test specimen, the length and the width of the straight-line segment were set to 20 mm and 10 mm, respectively. A universal tensile test machine (code number WAW–600 E) fabricated by Chuance Test Machine Co., Ltd. (Jinan, China) was adopted to test the static mechanical properties, as shown in Figure 6a. The estimated strain rate in the measuring section area was less than 0.0005 s^−1^ when the stretching speed was controlled at a velocity of 3 mm/min, fulfilling the regulations of the test standard noted in [27]. Therefore, the specimen was stretched with a velocity of 3 mm/min until fracture during the quasistatic tensile test. A Zwick HTM-16020 dynamic tension test machine was adopted to test the dynamic tension properties, as shown in Figure 6b, and serial tension tests were carried out with strain rates of 10 s^−1^, 100 s^−1^, 300 s^−1^ and 600 s^−1^.

Both for the quasistatic tensile test and dynamic tensile test, the nominal stress (also known as engineering stress) were calculated by dividing the tension force by the cross-section area of the test segment; similarly, the nominal strain (also known as engineering strain) was calculated by dividing the elongation length value by the original length of the test segment of the specimen. The calculation equations are expressed as *σ_nominal_* = *F*/(*w* × *t*) and *ε_nominal_* = Δ*L*/*L*, where *σ_nominal_* and *ε_nominal_* represent the nominal stress and nominal strain, respectively; *w* and *t* represent the width and thickness of the straight-line segment of the test specimens, respectively; and Δ*L* and *L* represent the elongation value and the original length of the test segment, respectively. Furthermore, in the quasistatic tensile test, the elongation was measured by a contact extensometer; in contrast, a non-contact optical measurement was adopted for the dynamic tensile test. The stress–strain relationships measured by both the quasistatic tensile test and the dynamic tensile test are shown in Figure 7. The mechanical properties of the mild steel obtained by quasistatic tension test are summarized in Table 1. Figure 7 shows that the dynamic yield stress increases with an increased in the dynamic strain rate, while fracture strain decreases with an increase in the strain rate. Since dynamic yield stresses are crucial for further investigations including theoretical solutions and finite-element analyses, the dynamic yield stresses under different strain rates are summarized in Table 2.

## 4. Results and Discussion

Dynamic responses of two types of structural specimens—conventional stiffened plates and strengthened stiffened plates—were tested in the present study. Two impact locations considered. Then, deformation modes, failure modes and the impact force–displacement relationships were compared and discussed.

### 4.1. Impact Locations

Two impact locations were considered. The first impact scenario is centric impact, which means that the impactor came into contacted with the stiffened plate at the center of the plate, as shown in Figure 8. For centric impact, the initial drop height of the impactor was set to 1.85 m, with an impact initial energy of 6555 J. Another impact scenario is eccentric impact, which means that the impactor came into contact with the stiffened plate eccentrically, with the impactor located on the plates between two stiffeners, as shown in Figure 9. With consideration of a lower load-carrying capability of the bare panel between two stiffeners, for the sake of safety, the initial drop height of the impactor was set to 1.2 m for eccentric impact, with an impact initial energy of 4259 J. Furthermore, both conventional plates and strengthened plates were investigated. Thus, four experimental tests were carried out in the present study.

### 4.2. Deformation and Failure Modes

In the centric impact scenario, the strengthening stiffener had a considerable influence on the deformation and failure mode of the stiffened plate under falling impact. For the conventional stiffened plate under impact, the panel underwent fracture, the stiffener under the impactor suffered from folding and the other stiffeners remained intact, as shown in Figure 10. For the strengthened stiffened plate, as shown in Figure 11, no obvious fracture occurred on the plate; meanwhile, the stiffeners did not suffer obvious inclined deformation—only bending deformation—due to the reinforcement of the extra stiffener.

However, the cross joint of the strengthened stiffened plate was a weak point. As shown in Figure 12, fracture occurred at the cross joint of the stiffeners on the strengthened plate under centric impact. The welding process led to a local welding stress concentration; furthermore, the material properties in the heat-affected zone tended to be brittle, which is the main reason for the initiation of cracks on the cross-joint area. In order to avoid undesired crack initiation and propagation, weld joints of the strengthened stiffener plate should be treated carefully.

In the eccentric impact scenario, the strengthening stiffener may significantly strengthen the load-carrying capability of the stiffened plate. Deformation and failure modes are shown for both stiffened plates in Figure 13; deformation and failure mainly occurred in the local area between the two stiffeners. As shown in Figure 13a, the plate was penetrated by the impactor due to a relatively low load-carrying capability of the bare plate. An I-shaped crack appeared in the plate because the crack was torn up at the two edges by the sharp impactor. In contrast, the strengthening stiffener enhanced the crashworthiness of the structure significantly. As shown in Figure 14, there only denting occurred on the plate, with obvious cracks the plate or the stiffeners.

### 4.3. Impact Force–Displacement Relationships

Impact force–displacement relationships indicate that the strengthening stiffener effectively improved the crashworthiness of the plate, as shown in Figure 15. Figure 15a presents the force–displacement responses of the stiffened plates under centric impact. For the conventional stiffened plates, the force reached the peak when the vertical displacement of the impactor reached approximately 22 mm, which means that the panel underwent fracture. For the strengthened stiffened plate, the force reached the peak when displacement of the impactor reached about 27 mm; then, the displacement of the impactor decreased, meaning that the impactor started to rebound. Figure 15b presents the force–displacement responses of the stiffened plates under eccentric impact. For the conventional stiffened plates, the force reached the peak when the vertical displacement of the impactor reached approximately 20 mm, meaning that the panel underwent fracture; then, the panel was torn by the impactor with increasing displacement. For the strengthened stiffened plate, the force reached the peak when displacement of the impactor reached about 26 mm; then, the displacement of the impactor decreased, meaning that the impactor started to rebound. Due to the strengthening of the stiffener, for centric and eccentric impact, the permanent plastic lateral deformation of the stiffened plate decreased by 20% and 26% respectively, showing that the crashworthiness of the stiffened plate was effectively strengthened by the extra stiffener.

## 5. Conclusions

In order to investigate the mechanical responses of deck structures of ships and offshore structures under falling-object impact, the dynamic behaviors of stiffened plates subjected to low-velocity impact of a wedge impactor were investigated in the present research. Useful conclusions are as follows:

(1) Dropping a wedge impactor caused catastrophic damage to the stiffener plate of deck structures. The sharp edge of the wedge impactor caused premature fracture of the plate, even under relative low impact energy.

(2) Stiffeners had positive influences on the impact responses of the deck structure. Under the same impact location, the permanent lateral deformation of the plate was reduced by approximately 20–26% due to the strengthening effect of a crossed strengthening stiffener. Furthermore, the strengthening stiffener avoided premature fracture of the panel, especially when impact was applied to the panel between two stiffeners.

(3) Residual stress and stress concentration of the cross-joint of the stiffeners, which are mainly caused by welding, may cause undesired fractures and should be treated carefully to avoid brittle fracture under impact.

## Figures and Tables

**Figure 1 materials-16-03128-f001:**
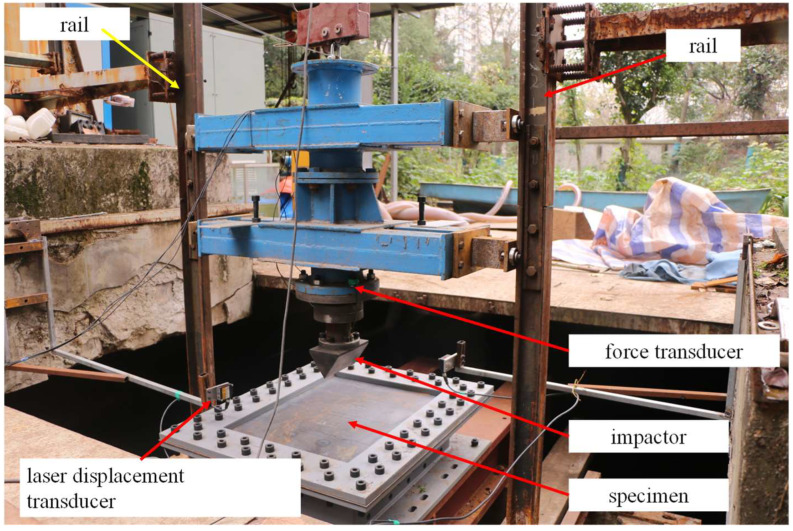
Drop-weight impact test tower.

**Figure 2 materials-16-03128-f002:**
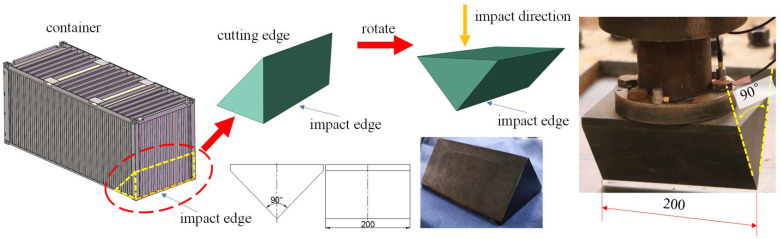
TEU (twenty feet equivalent unit) container and the rigid impactor.

**Figure 3 materials-16-03128-f003:**
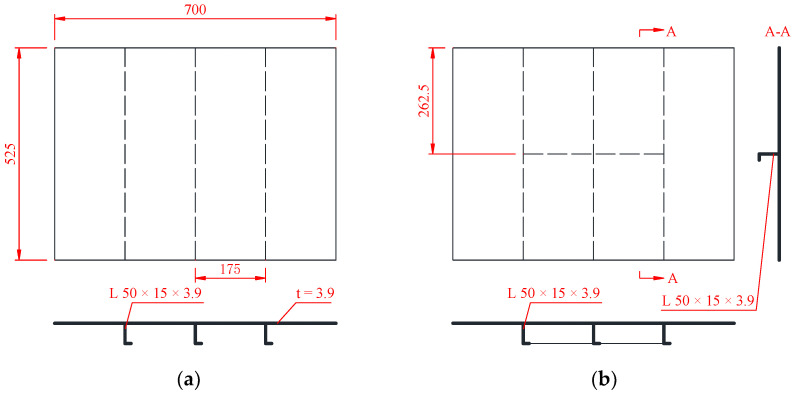
Geometric configurations of impact test specimens: (**a**) stiffened plate; (**b**) strengthened stiffened plate (unit: mm).

**Figure 4 materials-16-03128-f004:**
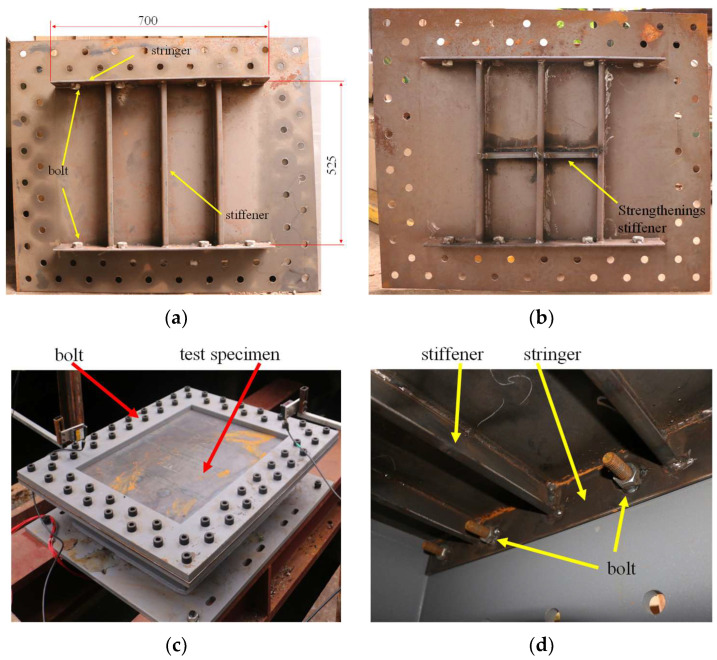
Photographs of the specimens: (**a**) conventional stiffened plate (unit: mm); (**b**) strengthened stiffened plate; (**c**) bolt arrangement; (**d**) stringer and the bolt on the backside.

**Figure 5 materials-16-03128-f005:**
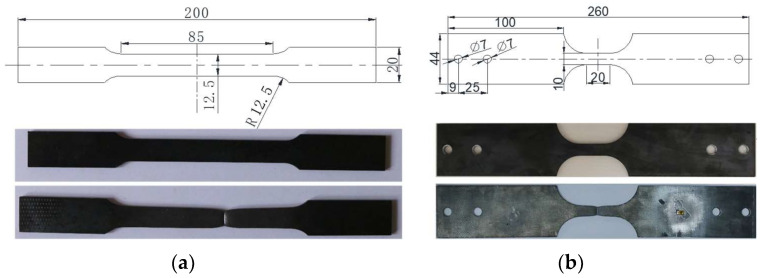
Dog-bone test specimens for (**a**) quasistatic tension and (**b**) dynamic tension (unit: mm).

**Figure 6 materials-16-03128-f006:**
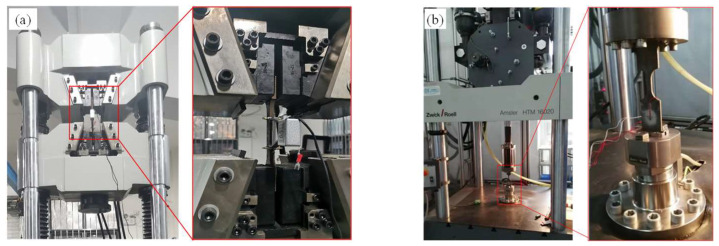
Test machines for unidirectional tensile tests of the material: (**a**) universal tensile test machine for quasistatic tension; (**b**) test machine for dynamic tensile test.

**Figure 7 materials-16-03128-f007:**
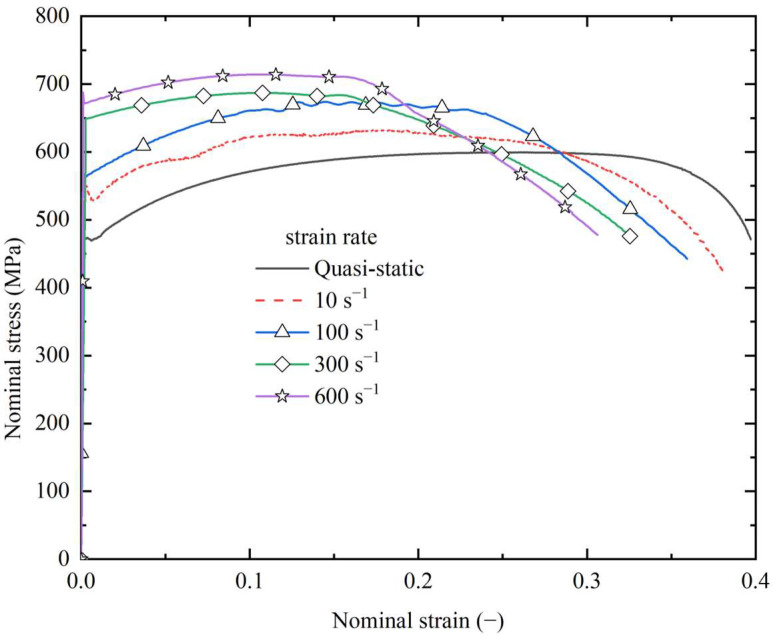
Tensile test results of the material.

**Figure 8 materials-16-03128-f008:**
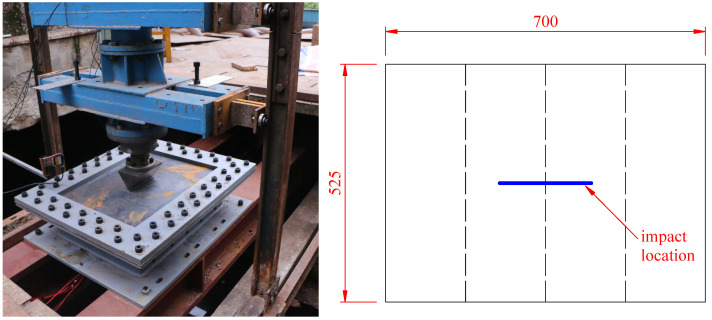
Centric impact scenario: impact at the center of the stiffened plate.

**Figure 9 materials-16-03128-f009:**
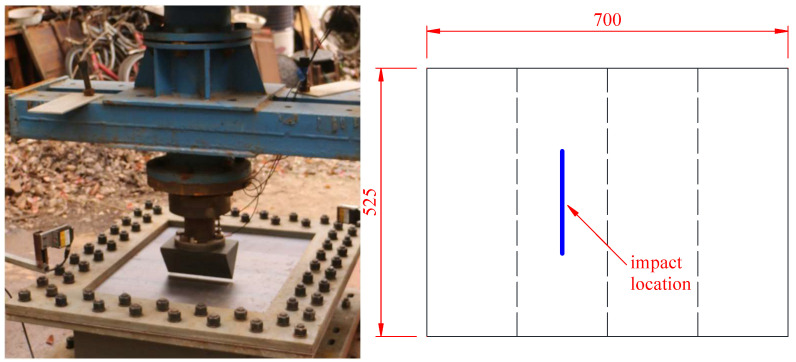
Eccentric impact scenario: impact on the plate between two stiffeners.

**Figure 10 materials-16-03128-f010:**
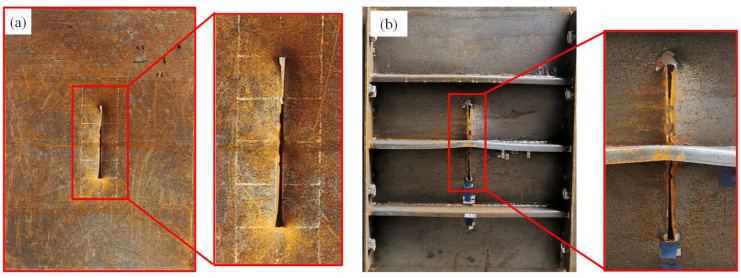
Deformation of the conventional stiffened plate under centric impact: (**a**) front side; (**b**) back side.

**Figure 11 materials-16-03128-f011:**
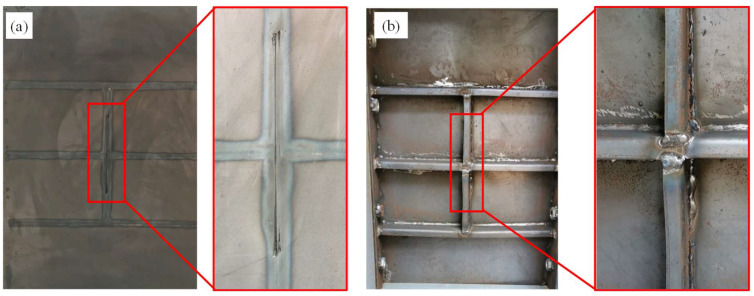
Deformation of the strengthened stiffened plate under centric impact: (**a**) front side; (**b**) back side.

**Figure 12 materials-16-03128-f012:**
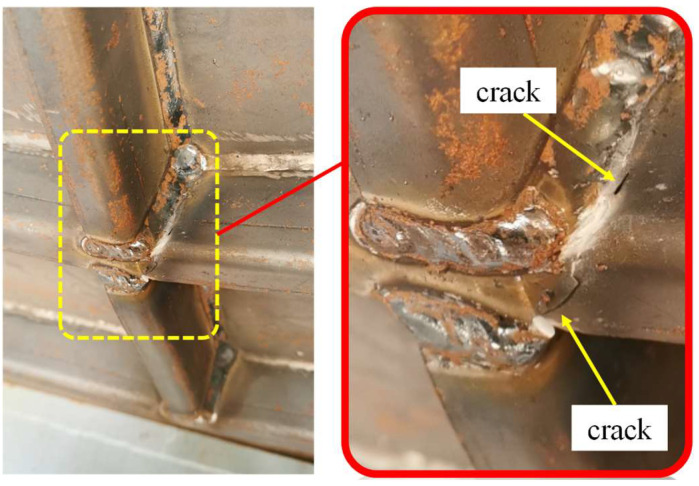
Fracture occurred at the cross joint of the stiffeners on the strengthened plate under centric impact.

**Figure 13 materials-16-03128-f013:**
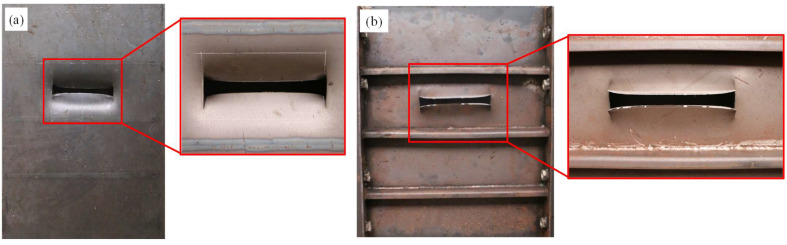
Deformation of the conventional plate under eccentric impact: (**a**) front side; (**b**) back side.

**Figure 14 materials-16-03128-f014:**
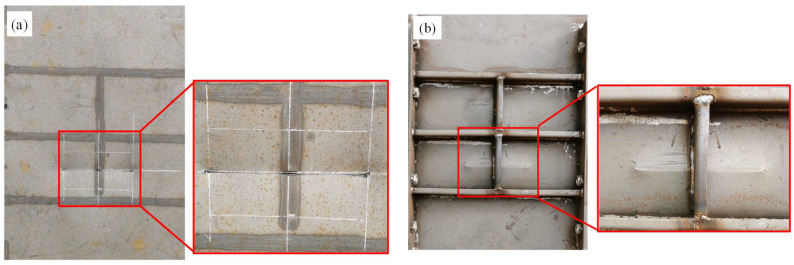
Deformation of the strengthened plate under eccentric impact: (**a**) front side; (**b**) back side.

**Figure 15 materials-16-03128-f015:**
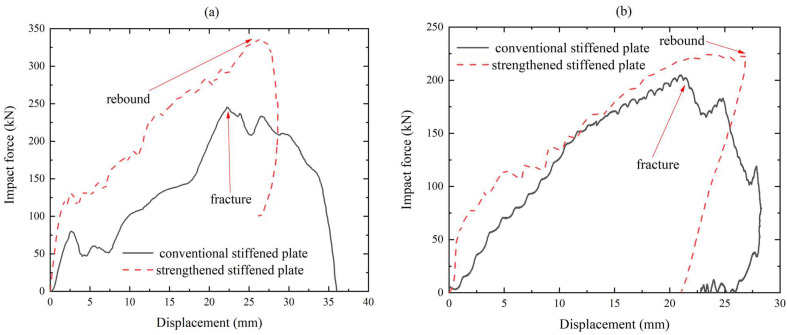
Comparison of impact force–displacement relationships: (**a**) centric impact; (**b**) eccentric impact.

**Table 1 materials-16-03128-t001:** Mechanical properties obtained by quasistatic tensile test.

Item	Mass Density	Elastic Modulus	Poisson’s Ratio	Yield Stress	Ultimate Tensile Stress	Fracture Strain
Units	kg/m^3^	GPa	-	MPa	MPa	-
Value	7850	210	0.3	472	786	0.33

**Table 2 materials-16-03128-t002:** Comparison of quasistatic and dynamic yield stress under various strain rates.

Strain rate (s^−1^)	Quasistatic	10	100	300	600
Yield stress (MPa)	472	530	561	649	675

## Data Availability

The data presented in this study are available upon request from the corresponding author.

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
