# Peer review of "Experimental Research on Dynamic Behavior of Stiffened Plates under Drop-Weight Impacts of a Wedge Impactor"

_materials, 2023, doi:10.3390/ma16083128_

Round 1
Reviewer 1 Report
Re-write lines 143-144. Speed of 3 mm/min is for quasi-static testing? Explain what is the "nominal stress" shown in Fig. 7.
Please check References (6, 9, 11, 13, 14) and write them in a consistent manner by following the indications given by the Editor.
Author Response
Authors acknowledge kind patience and useful advice from the reviewer.
The answers are listed below:
- Lines 143-144 in the original manuscript have been re-written. According to estimation, the strain rate in the measuring section area would be less than 0.0005 s^-1 when the stretching speed is controlled at a velocity of 3 mm/min, which would fulfill the regulations of the Chinese test standard.
- Nominal stress and nominal strain, also known as engineering stress and engineering strain, have been defined in section 3.2. Besides, the calculation method have also been added into the section 3.2.
- The references have been checked carefully with instruction from the Editor.
Reviewer 2 Report
The paper is well written. The stiffened plate structure is investigated under low velocity impact. It is observed that sharp wedge create more damage to stiffened plate. But due stiffener provide positive effect on the strength.
Author Response
Authors acknowledge kind patience and useful advice from the reviewer.
The extra stiffener have strengthened crashworthiness of the deck structure subjected to falling object impact. However, the mass properties of the deck structure have also increased.
Actually, the efficiency-cost ratio for the strengthening structure will be investigated in our schedule, and would be proposed in our following articles.
Reviewer 3 Report
The article presents good exposition about the experimental research of dynamic responses of the deck structure subjected to the impact of falling from a wedge-shaped pendulum. However, it needs to be improved.
1. In the first paragraph of the introduction, the text must be referenced.
2. From figure 2 it was not clear on which surface the impact will occur. According to the cut made in the container, there is no surface.
3. For a scientific nature, it was clear the relationship that the reinforcement welded on the surface generates during the impact. However, what is the technological nature to implement this manufacturing process in containers? What will be the extra weight, time and cost to accomplish this impact surface enhancement?
Author Response
Authors acknowledge kind patience and useful advice from the reviewer.
The answers are listed below:
- Authors add two references in the first paragraph of the introduction, to make the article more convictive.
- Figure 2 has been replaced with a new picture with more accuracy. In Fig 2, the impact edge of the TEU container have been identified clearly.
- The extra stiffener have strengthened crashworthiness of the deck structure subjected to falling object impact. However, the weight of the deck structure have also increased. Actually, the efficiency-cost ratio for the strengthening structure will be investigated in our schedule, and the corresponding results would be proposed in our following articles.
Reviewer 4 Report
Please, see the file attached.

Author Response
Authors acknowledge kind patience and useful advice from the reviewer.
The answers are listed below:
- Authors modify the literature review carefully with guidance from the reviewer. More literatures of experimental research on dynamic performances of ship structures under falling object impact have been cited in the first section.
- It is found that the gross weigth of the TEU container would be approximately 23 t. Meanwhile, the scaling factor of the deck structure is set to 4 in the present research. Thus, weight of the impactor in present research would be calculated by the scaling law: m=23000/4^3=359 kg. In the present case, the mass of the impactor was 362.2 kg, with acceptable accuracy. The corresponding text have been added into section 2.
- Dimensions in line 117-118 would be 700×525. Authors check the text carefully, and correct the clerical error.
- Authors list the main characteristics of the quasi-static and dynamic tensile test results in Table 1 and Table2, under the kind guidelines from the reviewer.
- Authors acknowledge reviewer for the useful advice. Actually, a statistical investigation with consideration of impact location and dimensions of stiffeners rely on a large amount of impact cases. It is too expensive and extravagant to carry out all these investigations by impact test. Thus the present manuscript provides two couples of typical impact tests to , and to give out a benchmark study for FEA investigation and theoretical solution. By following the kind instructions from the reviewer, we would propose further investigate by FEA, with consideration of impact location, impact angle, stiffener shapes influences, et al. The corresponding results would be proposed in our following articles.
- The purpose of the experimental study in the present manuscript is to explore the most basic phenomena and understand the deformation and damage mechanism of the deck structure under typical impact scenario. In our schedule, we will use the FEA to carry out further investigations with consideration of various structure configurations and mechanical properties. By then, useful conclusions for engineering application would be conducted and published in our following research articles.
Round 2
Reviewer 3 Report
I have read through the revised manuscript and, focusing on the changes made by the authors in response to the many comments and suggestions I offered in my original review. From my perspective, the authors have done a good job of revising the manuscript accordingly (and also responding to the other reviewers comments/questions). I judge the paper to suitable for publication in its current form and think it will become an interesting and valuable addition to the literature.
Reviewer 4 Report
The authors addressed almost all my comments and gave explanations when required. I have no further recommendations and wish them a fruitful research on the topic.